# Exploring the Effects of Energy Constraints on Performance, Body Composition, Endocrinological/Hematological Biomarkers, and Immune System among Athletes: An Overview of the Fasting State

**DOI:** 10.3390/nu14153197

**Published:** 2022-08-04

**Authors:** Hadi Nobari, Saber Saedmocheshi, Eugenia Murawska-Ciałowicz, Filipe Manuel Clemente, Katsuhiko Suzuki, Ana Filipa Silva

**Affiliations:** 1Faculty of Sport Sciences, University of Extremadura, 10003 Cáceres, Spain; 2Department of Exercise Physiology, Faculty of Educational Sciences and Psychology, University of Mohaghegh Ardabili, Ardabil 56199-11367, Iran; 3Department of Motor Performance, Faculty of Physical Education and Mountain Sports, Transilvania University of Braşov, 500068 Braşov, Romania; 4Department of Physical Education and Sport Sciences, Faculty of Humanities and Social Sciences, University of Kurdistan, Sanandaj 66177-15175, Iran; 5Physiology and Biochemistry Department, Wroclaw University of Health and Sport Sciences, 51-61 Wroclaw, Poland; 6Escola Superior Desporto e Lazer, Instituto Politécnico de Viana do Castelo, Rua Escola Industrial e Comercial de Nun’Álvares, 4900-347 Viana do Castelo, Portugal; 7Research Center in Sports Performance, Recreation, Innovation and Technology (SPRINT), 4960-320 Melgaço, Portugal; 8Instituto de Telecomunicações, Delegação da Covilhã, 1049-001 Lisboa, Portugal; 9Faculty of Sport Sciences, Waseda University, Tokorozawa 359-1192, Japan

**Keywords:** diet, intermittent fasting, calorie restriction, dehydration, exercise, balanced hormones, health promotion, hypoglycemia

## Abstract

The Ramadan fasting period (RFP) means abstaining from consuming food and/or beverages during certain hours of the day—from sunrise to sunset. Engaging in exercise and sports during the RFP leads to the lipolysis of adipose tissue and an increase in the breakdown of peripheral fat, leading to an increase in fat consumption. The effects of the RFP on functional, hematological, and metabolic parameters needs further study as existing studies have reported contradictory results. The differences in the results of various studies are due to the geographical characteristics of Muslim athletes, their specific diets, and their genetics, which explain these variations. In recent years, the attention of medical and sports researchers on the effects of the RFP and energy restrictions on bodily functions and athletic performance has increased significantly. Therefore, this brief article examines the effects of the RFP on the immune system, body composition, hematology, and the functionality of athletes during and after the RFP. We found that most sporting activities were performed during any time of the day without being affected by Ramadan fasting. Athletes were able to participate in their physical activities during fasting periods and saw few effects on their performance. Sleep and nutritional factors should be adjusted so that athletic performance is not impaired.

## 1. Introduction

Ramadan is one of the Islamic lunar months, and nearly 1.8 billion Muslims around the world fast during this month. During Ramadan, abstaining from eating and drinking can result in behavioral fluctuations and can have effects on athletes. During this month, Muslims avoid eating, drinking, and other activities that invalidate the fast, such as smoking and sexual intercourse. In terms of the spiritual aspect, it brings them closer to God. Fasting lasts from dawn to sunset for about 30 days, depending on the region’s geographical location [1]. Fasting during Ramadan can last from about 13 to 18 h, depending on the region where the Muslim resides. Ramadan and fasting can affect different body systems. The response of the immune system to fasting conditions is important because of its important role in regulating body homeostasis [1].

## 2. Energy Restriction

Controlling and maintaining weight is very important for athletes participating in competitions, especially in weight-based competitions [2]. Athletes should maintain their weight to be strong during competitions and prevent performance loss without a change in net muscle mass. Athletes often look for ways to lose weight quickly to reach their desired weights [3]. Exercise and physical activity cause weight loss by reducing body fat, along with reducing energy intake and energy consumption in sports. Studies have shown that regular exercise, along with optimal food consumption, improves the immune system [1,4].

One of the main principles of Islam, which is the fourth of the five pillars of faith, requires fasting during the holy month of Ramadan. During this time, Muslims abstain from eating, drinking, and engaging in sexual activity from before dawn until after sunset. Therefore, the duration of fasting and consequently the time for consuming food and liquids is determined by the times of the sunset and sunrise. This method of reducing energy intake is limited to the period between sunrise and sunset for 30 days. Muslims cannot consume any food due to their religious beliefs. During Ramadan, food diversity increases and one’s diet changes. For example, adolescent food consumption during Ramadan is largely associated with the increased consumption of fruits and vegetables and a diversity of foods. In addition, energy drinks, sweets, fats, and oils are consumed at a reduced rate. The consumption of cereal-based foods during Ramadan leads to a weight loss of about 1.5 kg during Ramadan [5]. There are various terms and expressions concerning energy limitations and the reduction of energy intake, some of which we will mention here [6].

One method of limiting energy and reducing energy intake that is most prevalent among Muslim athletes is an intermittent, or short-term, Ramadan fasting period (RFP), which involves the practice of not eating at certain times of the day beginning at sunrise [7]. Until sunset, one must then abstain from eating, consuming calories only at mealtimes. This method has features that have been more popular in different centuries of human history, especially among religious people [7,8].

Those who believe in the RFP and energy restriction usually consider this period as cleansing the body of waste products and resting the digestive system [9]. There are several ways to perform this method: an RFP and a short energy-restriction period (less than 24 h), an RFP and a moderate energy-restriction period (more than 24 h), and an RFP and a long energy-restriction period (more than 48 h) [10].

## 3. Fasting for All Athletes

In recent decades, one of the most interesting and popular topics among all people, especially athletes, has been weight loss. There are many interventions for weight loss, such as medications, diets, and exercise protocols. One such diet involves intermittent fasting (IF), which is one of the most up-to-date and common methods for weight loss. IF uses intervals of fasting and eating (non-fasting days or hours and fasting days). So far, there have been many studies performed on IF, showing different results [10,11]. It should be noted that IF is not a true diet; it is a method, a diet protocol requiring a time-consuming, planned approach, but it does not involve calorie restrictions. A person can fast by eating dinner early or by skipping an unplanned meal, so IF can be a part of normal life. Many factors affect the positive and negative aspects of IF. Depending on the goal, health status, and gender of the participant, there may be positive and negative implications [12]. Additionally, fasting is not starvation. While starvation can be a short- or long-term deprivation of food, fasting is a controlled deprivation of food for a certain period of time [13,14]. IF does not have a specific duration, and there are many protocols for its implementation, such as (i) shorter fasts: the 12:12, 16:8, and 20:4 models, in which the first number is related to the number of fasting hours and the second number is related to the number of eating hours, (ii) longer fasts: models involving fasting for more than 24 h, and (iii) even longer fasts: models involving fasting from 24 to 48 h [10].

## 4. Muslim Athletes and Fasting

The world’s total population includes 23% Muslims, and in about 50 countries, the majority of the population is Muslim [15]. Muslims follow religious laws in different ways, depending on their countries and religious beliefs [15]. However, the hours of the RFP vary depending on the latitude of a country. Usually, a summer RFP has different requirements than a winter Ramadan [16]. Training sessions occur during most months of the year, and international competitions can occur during Ramadan, requiring Muslim athletes to prepare for competitions and engage in training exercises while planning for RFPs [17]. Many athletes who compete in non-Muslim countries may encounter coaches who are not aware of these issues and may have problems with food planning [18]. Therefore, it may be difficult for instructors to plan exercise and nutrition programs for a number of people to meet their needs. Therefore, athletes and their coaches should pay attention to a few points. Eating and drinking water during Ramadan is unavoidable for Muslim athletes, but the athlete can arrange a training and competition program of his/her choice to plan for Ramadan [19]. Factors such as the type of sport and the need for training and competitions can affect Ramadan in Muslim athletes, and these athletes must plan their programs based on the characteristics of Ramadan [20].

## 5. Physiological and Metabolic Responses to an RFP and Energy Restrictions

The daily caloric requirement for women is 1600 to 2400, and for men, it is 2000 to 3000 calories, depending on age, size, height, lifestyle, health status, and level of physical activity [21]. Aging reduces calorie requirements due to a decrease in metabolism (1600 calories per day after age 51) [22]. The beneficial effects (Figure 1) of energy constraints are well-illustrated in the following diagram.

During an RFP, the level of insulin in the bloodstream is reduced, leading to glycogenesis and body-fat utilization [23]. Studies have shown that fat stores in the body decrease during food deprivation, a form of dietary restriction, in animal models [24] and human models [25]. Despite unconventional dietary restrictions, Muslim athletes engage in 1 month of IF during the year [24]. During fasting, the serum levels of ketogenic substances increase significantly, leading to the increased oxidation of fatty acids and reduced glucose energy consumption [24]. This process leads to the oxidation of excess fat and thus weight loss. Fasting also leads to alterations in the body’s lipid profile [24]. Ajibola et al. (2009) showed that long-term IF leads to a reduction in the serum levels of LDL cholesterol (bad cholesterol) and increased levels of HDL cholesterol (good cholesterol) during Ramadan. [24]. Cholesterol is excreted in two ways, by transfer to peripheral tissue by LDL and by transfer to the liver by HDL [26]. The body’s physiological response is to lower insulin levels during an RFP, dilate blood vessels, and release nutrients and oxygen to muscles and other organs to promote function [27,28]. Reducing or limiting energy intake during an RFP for 8 to 12 h causes the body to shift from glucose to fat for fuel. As the body turns to fat, the levels of catabolic hormones, such as growth hormone, cortisol, glucagon, adiponectin, and adrenaline, increase. These energy-induced changes cause blood glucose levels to rise [29]. This phenomenon usually occurs during RFP nights and sleeping periods. The disadvantages of energy constraints are well-illustrated in Figure 2.

## 6. Endocrine Adaptations Induced by an RFP

Limiting energy or engaging in an RFP causes hormonal changes in the body, such as changes in levels of thyroid hormones (reduction of T3) [30] and appetite-regulating hormones (decrease in hypothalamic orexin levels [26] and telencephalon NPY, CART, and CCK mRNA levels) [31] that regulate appetite and control body weight [32]. In RFP conditions and energy constraints, the body stimulates the thyroid gland to regulate energy by reducing blood flow [32], reducing exothermia [32], and lowering the basal metabolism [33]. Previous studies have shown that thyroid gland activity under these conditions increases by about 30% [28].

### 6.1. Hormones Released by the Thyroid and the Effect of an RFP on These Hormones

The thyroid gland is an endocrine gland located in the neck. It has two lobes that are separated by a narrow strip. These glands secrete the hormones thyroxine (T4) and triiodothyronine (T3) in response to TSH to control the metabolism of the body’s organs. Previous studies have shown that energy restriction and an RFP can reduce the circulating concentrations of thyroid hormones (T3 and T4) [34]. Studies have shown that fasting reduces serum T3 levels and increases serum reverse T3 (rT3) levels. This decrease in serum T3 is due to a decrease in the peripheral conversion of T4 to T3, which occurs in hepatocytes. Studies have shown that both serum T3 and rT3 levels return to pre-fasting values after breaking the fast. Similar results were observed for professional athletes during the RFP [30]. Even in obese and overweight people, the RFP slows down the hypothalamus–pituitary–thyroid pathway. The reason for this is the body’s reduction in energy consumption [35,36]. Usually, this decrease in thyroid hormones returns to pre-RFP levels after the 2 months of Ramadan.

### 6.2. Hormones Released by the Adrenal Gland and the Effect of an RFP on These Hormones

The adrenal glands are triangular glands located above the kidneys and are involved in regulating metabolism, blood pressure, and controlling the body during physical and mental stress. Due to RFP conditions, the body’s homeostasis is disrupted, causing stress in the body. One of the hormones secreted by the adrenal glands is cortisol, a steroidal hormone released in response to adrenocorticotropic hormone (ACTH). In response to stressful conditions, ACTH is secreted. One study found that sleep disorders and RFPs altered ACTH secretion, suggesting that cortisol levels increase under RFP conditions during Ramadan [37,38]. In one study, under normal circumstances, cortisol levels were increased in the afternoon compared to the morning. Changes in the sleep–wake cycle can increase morning values and produce a sharp drop afterward [39]. In regard to adrenaline and noradrenaline in RFP conditions, Zangeneh et al., (2015) observed that the serum levels of noradrenaline and adrenaline were significantly reduced compared to a control group [40].

### 6.3. Response of Appetite Hormones to an RFP

Appetite hormones (ghrelin and leptin) are usually very important during an RFP. These two hormones work in opposition to each other. Leptin increases appetite while ghrelin suppresses appetite [41,42]. In the RFP state, leptin levels increase in lean athletes [35,43,44], while after 3 weeks of an RFP, it decreases in men with the right body composition [45]. Studies have shown that leptin responds to both obesity and low nutrient conditions [46]. During energy restriction and an RFP, all organs of the body, including the genitals and reproductive organs, are affected [10]. For example, an 8-week RFP and energy restriction associated with strenuous exercise decreased serum testosterone levels in lean, healthy men [47,48]. The physiologic properties of testosterone stimulate protein anabolism causing hypertrophy [49], which is reduced during energy restriction [50]. During energy restriction, the level of insulin-like growth factor-1 (IGF-1) decreases [51], but this has no significant effect on growth hormone (GH) [52].

## 7. Nutritional Considerations for Athletes during an RFP

Macronutrients are very important for athletes. During an RFP and energy restriction, athletes lose weight due to reduced food intake. Therefore, a proper diet plan during this period can help an athlete achieve a good balance between energy intake and consumption [53,54]. However, athletes should not focus on micronutrients, but rather they should include macronutrients in their diets [55].

### 7.1. Protein

Consuming protein is one weight management strategy that increases the feeling of satiety as well as heat production [56,57]. People with high-protein diets do not have a big problem because they are used to consuming less energy [58,59]. Therefore, planning a high-protein diet can be a solution for athletes during RFPs. High-protein diets increase satiety, increase thermal energy, and reduce fat consumption [12]. Using protein supplements, along with resistance training and other supplements, such as creatine and beta-hydroxy-beta-methylbutyric acid (HMB), helps maintain energy [19,60].

### 7.2. Lipids

Adequate lipids in the diet are very important due to the structural and storage properties of this nutrient [61]. Therefore, engaging in resistance training during energy restriction can induce the consumption of carbohydrates and lipids, and it can eliminate the consequences of an RFP. The dietary recommendation for RFPs is to consume a low-fat meal with a high carbohydrate level [62,63,64]. It has been suggested that fasting increases the number of beta-adrenoceptors in adipose tissue in rats [65], but a decrease in beta-adrenoceptors has been observed in human adipose tissues [65]. Fasting conditions lead to an increase in the basal lipolysis of adipose tissue, but the lipolytic response (even of antihypertensives) to catecholamines decreases [66]. Fasting increases the levels of catabolic hormones, such as GH [67] and glucagon [66], as well as catecholamines, which leads to increased lipolysis [68].

### 7.3. Hydration Strategies

Maintaining a water–salt balance during an RFP is important for athletes, but it is challenging when they must continue their training and competition programs [69,70,71]. A lack of water can create three conditions for Muslim athletes:⮚The RFP prevents athletes from being hydrated during a race, and they become dehydrated.⮚The lack of water causes a drop in performance.⮚The lack of water disrupts the balance of water and electrolytes [71].

Many factors affect exercise performance during RFP conditions [71]. The duration of exercise is an important factor as short-term activities are less affected by RFP conditions. As the duration of food and water deprivation increases during Ramadan, so does the condition of hypohydration [7,72]. It has been observed that the deprivation of fluids reduces one’s level of consciousness and results in mental fatigue, ultimately causing functional impairment [73,74]. However, ambient temperatures and humidity can affect the hydration of the athlete [75,76]. Many criteria can be used to diagnose dehydration, such as the frequency, volume, and color of the urine [77]. Using the following solutions can prevent dehydration in the body during an RFP:Reduce the time in situations where there is a possibility of dehydration, such as exposure to sunlight.Be sure to check the hydration status of the body.Before an RFP begins, fluids should be selected according to the conditions and contain minerals and energy that reduce water loss.The consumption of fluids when a person is not participating in an RFP should be regular, such as during the night.As much as possible, one should train and compete for 2 to 3 h after sunset [78].

## 8. Response of Hematocrit to an RFP in Athletes

During 1 of the 12 months of the Islamic calendar (“Ramadan”), mature and healthy Muslims abstain from drinking and eating from sunrise to sunset [78]. An RFP during the holy month of Ramadan can cause many changes in the psychological and physiological characteristics of a person [79,80]. Physiological changes that have been observed include dehydration [79], metabolic responses [79,80], sleep and wakefulness changes, hormone secretion, and the function of various organs of the body [80]. Athletes must manage these changes by developing a proper training plan [81]. By considering one’s water supply and nutrition during Ramadan, the stability of the athlete’s blood counts can be maintained [82].

### 8.1. Erythrocyte States

Studies have shown conflicting results regarding Ramadan’s effect on the body’s iron status. Boehlell et al. [83] observed a significant increase in hematocrit (Hct) and hemoglobin (Hb) concentrations in rugby players at the end of Ramadan compared to a control. However, Hosseini and Hejazi [84] observed a decrease in Hct, Hb, and red blood cells (RBC) in young football players following an RFP. Additionally, Hosseini et al. [85] failed to observe significant changes after a Ramadan RFP in Hct and Hb but saw a decrease in RBC [85]. When considering different times of the day, Maughan et al. [86] reported small changes (less than 2%) in the Hct levels of football players during Ramadan.

### 8.2. Platelet Count

When examining the effect of an RFP on blood platelet count in three sports, an increase in the number of cells was seen in wrestlers [84], but this increase was not observed in football players [87] or weightlifters [88].

## 9. Immunosuppression and Related Cells during an RFP

Malnutrition is a dangerous factor affecting the homeostasis of the body and the maintenance of the physiological balance of living organisms. Usually, all living organisms have a way to control their energy reserves. When energy availability is high, the body stores the excess energy in fat tissue, and when the energy availability is low, the fat tissue is used. Malnutrition and overnutrition can cause cytokine and hormone responses to these conditions. One of the first reports regarding this included tumor necrosis factor-alpha (TNF-α). Overfeeding was found to increase levels of TNF-α, whereas undernutrition caused a decrease in TNF-α levels [89]. TNF-α is a known pro-inflammatory cytokine essential for the acute phase response. Studies have shown that reducing energy intake can suppress the immune system and increase the incidence of infectious diseases [90]. Figure 3 shows the impact of energy deprivation and malnutrition on the immune system.

## 10. Response of the Immune System to an RFP in Athletes

An RFP significantly reduces inflammation and reduces the incidence of cancer in animal models [91]. In a human study (with 21 males and 29 females), changes in circulating pro-inflammatory cytokines, such as interleukin 1β (IL-1β), interleukin 6 (IL-6), and tumor necrosis factor (TNF), were evaluated [92]. TNF-α and immune cells (total leukocytes, monocytes, granulocytes, and lymphocytes) were observed during 3 periods: 1 week before Ramadan, 3 weeks after Ramadan, and 1 month after Ramadan. It was observed that the pro-inflammatory cytokines IL-1β, IL-6, and TNF-α decreased significantly during Ramadan [93]. In addition, a significant decrease in immune cells was observed during Ramadan, but they returned to baseline [94]. The effect of Ramadan on oxidative stress and cell damage in healthy individuals was evaluated in 14 healthy volunteers. Researchers observed no significant changes in oxidative stress by measuring malondialdehyde (MDA), glutathione [95], glutathione peroxidase [95,96], and catalase levels after Ramadan [97]. An RFP decreased IL-12 levels during weeks 1 and 4 of Ramadan as compared to week 3. Changes in IL-12 levels can be a result of changes in diet and sleep patterns [98]. It has been observed that an RFP can increase macrophage levels and decrease bacterial levels in the body. In addition, this study showed that an RFP can increase interferon-gamma (IFN-γ), which activates antimicrobial mechanisms in the body [99]. The chart in Figure 4 clearly shows the impact of Ramadan on the immune system.

## 11. Changing the Performance and Record of Athletes during Ramadan

Many studies have examined the effects of energy restriction and RFPs on physical activity [100], but most of these studies included beginner athletes [101]. Few studies have examined the effect of RFPs and energy constraints on the performance of professional athletes [102,103,104]. The positive effects of physical exercise and an RFP on the body and overall health have been proven. Both are positive ways to increase the lipolysis of adipose tissue and muscle and reduce body fat [105]. This is also very important for Muslim athletes, as they take care of their bodies and try to maintain and improve their athletic performances [105]. It is important for Muslim athletes to be in an RFP state and exercise as it is important for them to perform their religious duties and maintain and strengthen their performance and training record [105]. When examining the effect of energy restriction and short-term RFPs on exercise performance, it was observed that physical function decreased, and this decrease was evident during an RFP. This result has been observed during moderate-term RFPs (24 to 55 h), and its main causes have been associated with dehydration; prolonged, tedious exercise; and/or very high levels of exercise. Contrary to these findings, researchers did not observe a significant reduction in athletic performance during short RFPs (11 to 24 h) [106,107]. For example, studies by Fashi et al. [104] and Stannard et al. [108] found that even a short fast can have a positive effect on a muscle’s physiological adaptation to endurance exercises. Of course, this research was performed on beginners and non-professionals. The changes observed in endurance performance during Ramadan are not limited to the training protocol, and the physiological adjustments resulting from an RFP can lead to a slowing of the metabolism during these conditions [109]. In the first week of an RFP, serum sodium, chloride, and protein levels usually increase (during the dehydrated period), a process that results in a loss of 1.13 kg of body weight, which is mostly due to dehydration. Following an increase in these factors, the release of catecholamines is suppressed, causing less venous blood to return to the heart, ultimately leading to a decrease in sympathetic function. An RFP also reduced functional capacity and maximal oxygen consumption. These changes returned to their original state when the fasting was over [109,110].

### 11.1. Endurance Training

Aragon-Vargas [111] reviewed the effects of an RFP lasting from 24 h to 4 days on endurance exercise and concluded that it had a negative effect on endurance. Most studies on RFPs have looked at the short-term effects, with few studies evaluating the long-term effects of RFPs. According to the literature, it takes 10 days to adjust to an RFP. In addition, most studies have examined the effect of an RFP on animal models [112,113]. Studies on animal models are difficult to correlate to humans. Zerguini et al. [114] observed a decrease in physical performance in Algerian players during Ramadan.

### 11.2. Resistance Training

The reduced energy-intake during an RFP can lead to a decrease in lean mass and strength, which could reduce one’s production of force and power [115]. Real-Hohn et al. [116] evaluated a fasting period (20:4) for 4 days a week and did not observe a significant effect on muscle production and muscle cross-section. Several other studies have not observed a significant effect of RFPs on energy production or power. Trabelsi et al. [117] examined the effect of an RFP on bodybuilders and did not observe a significant difference in body mass or body composition.

### 11.3. Sprint Performance

Various studies have shown a decrease in anaerobic capacity during Ramadan. Salama et al. [118] showed that an RFP reduced aerobic performances in the 100- and 800-m races. However, many studies [7,119] observed only a slight decrease in the mean sprint performances, but these findings were not statistically significant. Additionally, by keeping the training load constant during Ramadan, sprint speeds were significantly faster [120].

### 11.4. Power-Output Measure

Studies have shown that an RFP had no significant effect on a participant’s performance on Wingate anaerobic tests (peak power and average power) [121,122] but they showed a significant reduction in anaerobic capacity compared to the non-RFP hours of 17:00 and 21:00 [122].

### 11.5. Blood Lactate Concentration

RFPs did not have a significant effect on lactate accumulation and the time required to reach maximum blood lactate levels [104,123]. Furthermore, researchers did not observe a significant effect on blood lactate concentrations due to a 20-min run during Ramadan [124]. Additionally, another study examining the 30-s vertical jump test in judo did not observe significant changes [101].

Conflicting results have been observed in many studies. There are several reasons for such differences, including the length of fasting and the physical characteristics of the study participants. According to various studies, athletes should be advised to remain moderately active during Ramadan to ensure adequate performance and energy storage. It is recommended that more studies be performed on fasting status and athletic performance. These studies should evaluate the different mechanisms the body uses to respond to fasting conditions during Ramadan and compare them to the results of Ramadan fasting on the physiologic variables of athletes. Table 1 summarizes the studies evaluating the effects of fasting conditions on exercise performance and physiologic conditions of the body.

### 11.6. Practical Applications

This review article sought to provide information about Ramadan and the performance of athletes. The main results showed that the average short-term performance during the month of Ramadan is reduced in the afternoon. To date, most Muslim athletes have a belief-based view of Ramadan. The results of various studies showed that Ramadan does not have a detrimental effect on performance. Therefore, it seems that athletes may be able to compete on an empty stomach without reducing their physical function. Priority is given to improving sleep and nutrition, which can have an impact on performance from sunrise to sunset.

## 12. Conclusions

This review article attempted to examine the effects of an RFP on exercise performance and the physiology of the body. An RFP and fasting leads to improvements in many cases. Muslim athletes should plan their training and competition programs based on the duration of the fast so that their performance is not impaired. Athletes can advance their training programs by manipulating training variables, such as the intensity, duration, volume, and frequency of training, during Ramadan.

## Figures and Tables

**Figure 1 nutrients-14-03197-f001:**
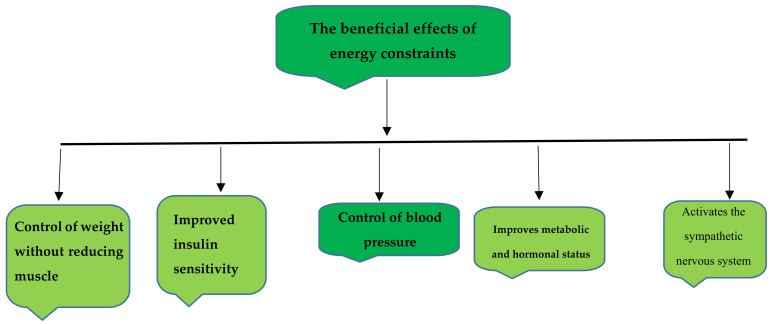
The beneficial effects of energy constraints on the body.

**Figure 2 nutrients-14-03197-f002:**
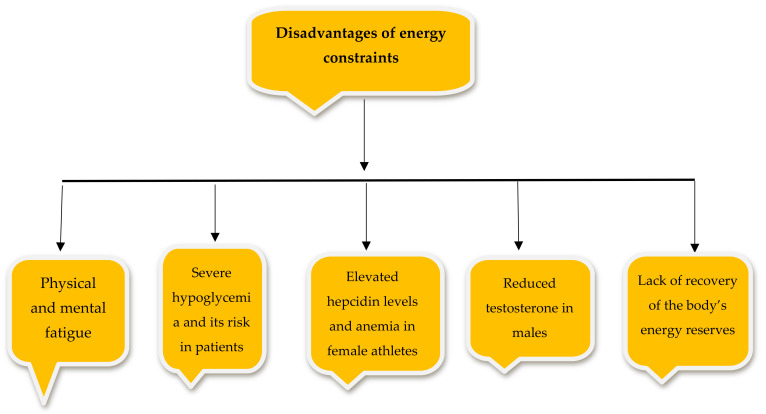
The disadvantages of energy constraints on the body.

**Figure 3 nutrients-14-03197-f003:**
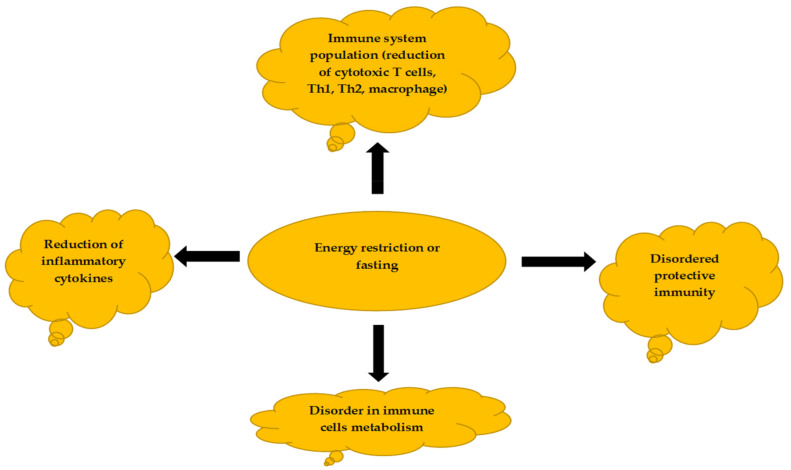
Changes in energy availability can drastically impact immune cell functions.

**Figure 4 nutrients-14-03197-f004:**
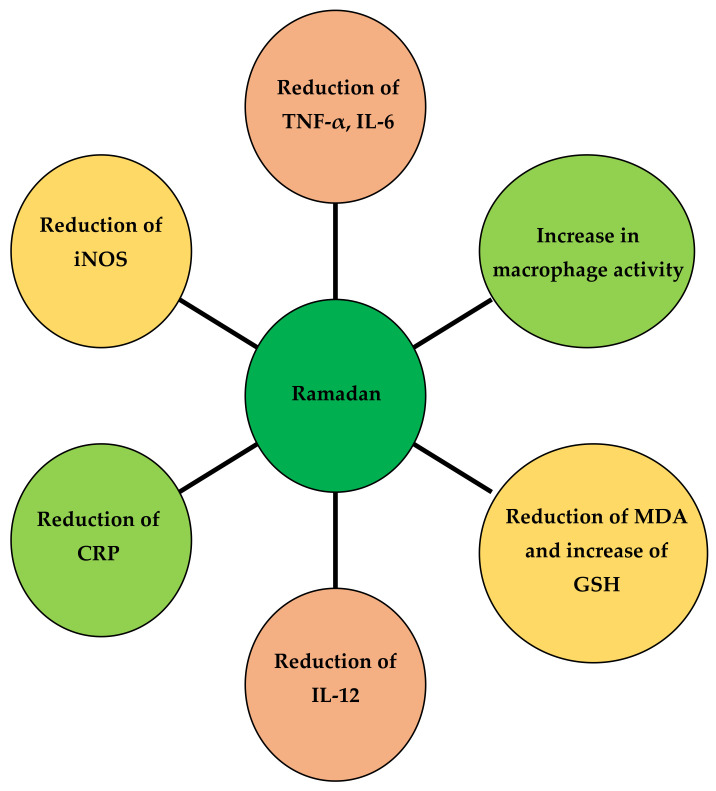
The beneficial effects of energy constraints on the immune system.

**Table 1 nutrients-14-03197-t001:** Fasting, exercise training, and the body’s response.

Authors	Sample	Protocol	Results
Attarzadeh Hosseini et al. [84]	26 healthy males (two experimental groups were compared before and after the training period).	Participants were divided into non-active fasting (n = 13) and active fasting (n = 13) groups.	Positive alterations in hematological–biochemical Indices (Hb and Hct decreased; plasma glucose reduced significantly).
Dewanti et al. [125]	100 male outdoor workers.	Before the start of Ramadan and during the third week of the month of Ramadan.	Blood pressure was reduced in the partial-fasting and non-fasting groups, which was an unexpected result. While red blood cell production was suppressed, as evidenced by lower levels of Hb, red blood cells (RBC), and packed cell volume (PCV), the subjects were normocytic and normochromic based on normal mean corpuscular volume (MCV), mean corpuscular hemoglobin (MCH), and mean corpuscular hemoglobin concentration (MCHC) levels.
Chaouachi et al. [94]	15 elite male judo athletes.	Before, during, and after Ramadan (maintaining their usual high training loads).	The RFP produced small but significant changes in the inflammatory, hormonal, and immunological profiles of the judo athletes. Serum C-reactive protein increased from 2.93 ± 0.26 mg·L^−1^ pre-Ramadan to 4.60 ± 0.51 mg·L^−1^ at the end of Ramadan. Haptoglobin and antitrypsin significantly increased during different phases of Ramadan, whereas homocysteine and prealbumin levels remained relatively unchanged. Albumin decreased slightly by mid-Ramadan, then recovered. Immunoglobulin A increased from 1.87 ± 0.56 g·L^−1^ before Ramadan to 2.49 ± 0.75 g·L^−1^ at the end and remained high for 3 weeks after it ended. There were no changes in leucocyte cell counts throughout the study. The mean blood levels of thyroid-stimulating hormone and free thyroxine increased significantly during the RFP.
Basilio et al. [126]	Wistar rats (n = 60) were randomly divided into 4 groups: control, exercise training (ET), intermittent fasting (IF), and exercise training plus intermittent fasting (ETI).	Over 12 weeks, control and ET animals were fed a standard, commercial diet ad libitum daily, while IF and ETI animals were fed every other day. In addition, the ET and ETI groups were submitted to a running protocol on a treadmill.	Exercise training increased the functional fitness of the ET and ETI groups and promoted cardiac fibrosis. The combination of IF and exercise training resulted in a smaller area under the blood-glucose curve and reduced the cardiomyocyte cross-sectional area and the interstitial collagen fraction in the ETI group as compared to the ET group. ERK and JNK expression levels were similar among groups (*p* > 0.05).
Schübel et al. [127]	150 overweight and obese participants (50% males, 50% females), age 35–65, with BMIs between 25–40 kg/m^2^ were divided into three groups (CCR: n = 49, ICR: n = 49, CG: n = 52).	→ICR: 5 days without energy restriction and 2 days with 75% ↓ in energy needs. → CCR: 20% daily ↓ in energy needs. →CG: NC in calorie intake. Over a 12-week intervention phase, a 12-week maintenance phase, and a 26-week follow-up phase.	Body weight during the intervention phase decreased by 7.1% ± 0.7% (*p* < 0.001) in the ICR group, by 5.2% ± 0.6% (*p* = 0.053) in the CCR group, and by 3.3% ± 0.6% (NS) in the CG group. At the final follow-up assessment (week 50), weight loss was 5.2% ± 1.2 (*p* = 0.01) in the ICR group, 4.9% ± 1.1% (*p* = 0.89) in the CCR group, and 1.7% ± 0.8% in the CG group.

↓: Reduction of the desired variable.

## Data Availability

Not applicable.

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
