# Peer review of "Exploring the Effects of Energy Constraints on Performance, Body Composition, Endocrinological/Hematological Biomarkers, and Immune System among Athletes: An Overview of the Fasting State"

_nutrients, 2022, doi:10.3390/nu14153197_

Round 1

Reviewer 1 Report

In this review paper the authors aimed to discuss the Ramadan fasting period and immunological alterations. 
The manuscript is well written and presented, and the results provided interesting information to that may be relevant for knowledge in terms of physiological and immune adaptation and also for proper training management.
However, I believe the authors need to address some points before making strong conclusions.

Changes in the energetic available might promote drastic impact to immune cells functions. Recent review papers have documented this scenario. On the other hand, long time of the poor energetic environment might leads to immunosuppression status (Clin Nutr. 2003 Oct;22(5):453-7). Please, this point will be excellent to improve the present discussion. 

Another point that authors can insert in discussion is related with immunosuppression and naive cells during RFP. 

Author Response

Response to reviewers: Manuscript ID: nutrients-1817006

Reviewer #1

Dear Reviewer,

We have carefully considered all considerations in the document provided by you. Enclosed you will find our detailed answers to your inquiries.

Thank you for the time taken to review our paper and for giving us the chance to improve it.

We respond point by point below.

Reviewer 1

In this review paper the authors aimed to discuss the Ramadan fasting period and immunological alterations. 
The manuscript is well written and presented, and the results provided interesting information to that may be relevant for knowledge in terms of physiological and immune adaptation and also for proper training management.
However, I believe the authors need to address some points before making strong conclusions.

Changes in the energetic available might promote drastic impact to immune cells functions. Recent review papers have documented this scenario. On the other hand, long time of the poor energetic environment might leads to immunosuppression status (Clin Nutr. 2003 Oct;22(5):453-7). Please, this point will be excellent to improve the present discussion. Another point that authors can insert in discussion is related with immunosuppression and naive cells during RFP. 

Response: Thank you for your valuable comment. You mentioned a very important point that the authors applied in the article and added the effect of restriction and low energy environment on the immune system as a subsection, the names are below.

  • Energy restriction
  • Immunosuppression and related cells during RFP
  • Figure 3

Reviewer 2 Report

Dear authors,

the topic is interesting, however I found the review difficult to read and follow.

Specifically, the following issues need to be addressed:

- title: it does not mention Ramandan fasting period (RFP), the focus of the research

- text and literature search: literature research mixes studies performed on non-athletes with studies on athletes, so it is very difficult to follow and draw conclusions

- obesity is mentioned as a clinical burden (truely stated) but it seems not to be an issue for athletes, so if the review focuses on athletes, probably it's not that relevant 

Author Response

Response to reviewers: Manuscript ID: nutrients-1817006

Reviewer #2

Dear Reviewer,

We have carefully considered all considerations in the document provided by you. Enclosed you will find our detailed answers to your inquiries.

Thank you for the time taken to review our paper and for giving us the chance to improve it.

We respond point by point below.

Reviewer 2

the topic is interesting, however I found the review difficult to read and follow.

Specifically, the following issues need to be addressed:

- title: it does not mention Ramandan fasting period (RFP), the focus of the research

Response: We added to text and part of introduction revised title and focus more on RFP.

- text and literature search: literature research mixes studies performed on non-athletes with studies on athletes, so it is very difficult to follow and draw conclusions

Response: Thank you for your comment, the part of non-athletes and obese people was removed from the article and only athletes were discussed.

- obesity is mentioned as a clinical burden (truely stated) but it seems not to be an issue for athletes, so if the review focuses on athletes, probably it's not that relevant 

Response: We applied your comment in the article and removed the non-athletes and obesity section from the article and only discussed Ramadan and athletes.

Round 2

Reviewer 2 Report

Dear authors,

I appreciated the improvements you made to the manuscript. However, I ask you to revise your text, as the English writing need to be improved.

Author Response

Dear Reviewer,

The article has been completely edited by a native person. You can see all the changes with "track changes" in the article.

All the best,

Hadi Nobari (on behalf of authors)